# Mapping retracted articles and exploring regional differences in China, 2012–2023

**Liping Shi**[1☯], **Xue Zhang**[1☯], **Xiaojun Ma**[1], **Xian Sun**[1,2], **Jiangping Li**[1,2]*, **Shulan He**[1,2]*

1 Department of Epidemiology and Health Statistics, School of Public Health, Ningxia Medical University, Yinchuan, Ningxia, China, 2 Key Laboratory of Environmental Factors and Chronic Disease Control, Ningxia Medical University, Yinchuan, Ningxia, China

☯ These authors contributed equally to this work.
* lijp@nxmu.edu.cn (JL); heshulan0954@163.com (SH)

**Data Availability Statement:** The original dataset can be downloaded from the Figshare public repository (https://doi.org/10.6084/m9.figshare.27044899).

## Abstract

### Background

China is one of the top countries with the most significant number or proportion of retracted publications, which has garnered significant attention.

### Methods

Using the Retraction Watch Database, we collected retracted articles written by Chinese authors from 31 provinces in mainland China, spanning the period between January 1, 2012, and December 31, 2023. We used Geographical Information Science to analyze spatial distribution characteristics of retracted articles by Chinese authors and identify high-risk clusters of retracted areas.

### Results

A total of 14,445 retracted articles authored by researchers from 31 provinces in China between 2012 and 2023 were analyzed. The Spatial trend surface analysis and Gravity center movement indicated a gradual increase in the number of retracted articles from the west to the east. The spatial autocorrelation analysis showed that revealed significant spatial clustering in the distribution of retracted articles across the 31 provinces. The results of the spatial-temporal clustering analysis showed that the hotspots were primarily concentrated in Shandong, Jiangsu, Shanghai, Henan, and Anhui.

### Conclusion

There is a discernible spatial clustering among these retractions, with a gradual increase in the number of retracted articles from west to east. Shandong, Jiangsu, Shanghai, Henan, and Anhui are the hotspots for retractions.

**Funding:** The author(s) received no specific funding for this work.

**Competing interests:** The authors have declared that no competing interests exist.

## Introduction

Scientific research helps to promote innovation and provide an effective model for development [1]. However, academic misconduct and inaccuracies in scholarly publications undermine academic progress and erode public trust in science. Retraction, according to the definition of the Committee on Publication Ethics (COPE), is a mechanism for correcting academic literature by alerting readers to publications that contain flawed or erroneous data [2]. Therefore, retracting questionable articles constitutes a vital act of social responsibility within the scientific community [3]. In recent years, the number of retracted articles has continuously increased. *Nature's* analysis suggests that in 2022 the retraction rate exceeded 0.2%, and more than 10,000 research articles were retracted in 2023 [4]. As an aspect of science [5], the main objectives of retraction are to avoid invalid conclusions that could lead to misguided patient management, preserve the trustworthiness of scientific literature, and eliminate from the scientific environment [6, 7]. Retractions not only squander valuable time, human labor, and research resources, but also damage the reputation of the researchers and hinder their professional growth [8]. A previous study assessed that the cost of a single research paper was more than 90,000 RMB, based on the data of the National Natural Science Foundation of China from 2006 to 2016 [9]. In addition, the reputation and interests of the journal are also affected. Matthew Kissner, Wiley's interim chief executive, said that the publisher expects to lose between US$35 million and $40 million in revenue this fiscal year because of Hindawi's paper mill problem [4]. Numerous reasons for retractions include fraud (data fabrication or manipulation), lack of appropriate ethical approval, duplicate publications, plagiarism, and studies with methodological or other nonfraud data issues [10].

China is an important producer of scientific articles and has become a major contributor to global scientific research [11]. Previous studies show that China is one of the countries with the highest number or proportion of retracted publications [4, 12–15]. Research misconduct in Chinese academia has attracted international attention. According to *Nature*, Hindawi issued "more than 9,600 retraction notices in 2023, of which about 8,200 had Chinese co-authors", and "nearly 14,000 retraction notices were issued by all publishers in 2023, with about three-quarters involving Chinese co-authors" [16]. This harms China's academic reputation and academic environment. Thus, it is essential to focus on how to prevent and address the retraction phenomenon.

The number of retracted articles is closely related to the amount of research and the degree of development in the region [17]. Geographic Information Systems (GIS) is a computer technology and software for collecting, managing, and analyzing spatial data. It can effectively use spatial information to simulate and analyze historical surveillance data and obtain statistically significant clusters. Additionally, GIS visualizes spatially distributed data for use in a wide range of applications [18], such as spatial econometrics, ecology, and biology [19–21]. The introduction of GIS technology has changed the traditional analysis concept [22]. By using spatial analysis technology and considering spatial autocorrelation, the geospatial distribution pattern of the number of retracted articles can be revealed and the aggregated area of retracted articles can be identified. Quantitative analysis of retracted articles, analysis of reasons for retraction, the citation mode of retracted articles, strategies to deal with retracted articles, publishing ethics, and the construction of scientific research integrity are often the most salient aspects of empirical research on retracted publications in China [23, 24]. However, few empirical studies have used spatial analysis techniques to explore the relationship between Chinese authors' number of retracted articles and regions.

Therefore, the purpose of this study is to analyze the spatial distribution characteristics of the number of retracted articles by Chinese authors from 31 provinces across the country in 2012–2023. This will identify high-risk clusters of retraction areas, as well as provide insights for adopting more targeted measures to prevent academic misconduct and to improve the integrity of the scientific research system in specific regions of China.

## Methods

### Data source and inclusion criteria

We utillized the Retraction Watch Database for our study [25] (http://retractiondatabase.org/). Officially launched in 2018, the Retraction Watch Database is the most comprehensive index of retractions of published articles, encompassing notices of retraction spanning from 1753 to the present, and including author, title, the reason for retraction, subject, journal, publisher, affiliation, notes, and so on. The subjects included Business & Technology, Environmental Sciences, Health Sciences, Physical Sciences, Basic Life Sciences, Social Sciences, and Humanities. Retraction records and notices are collected daily through email alerts, user submissions, and thorough examination of journal contents, as well as utilizing various databases including PubMed and Google Scholar, along with publisher sites such as Springer Link and Science Direct. The Retraction Watch Database is also referenced for retraction data, which has been employed in numerous studies [26–28].

The records downloaded from Retraction Watch Database were filtered by the country "China" and included only those with a retraction date between January 1, 2012, and December 31, 2023. This study focuses solely on the institutions of the first authors of the retracted articles from the 31 provinces in mainland China. The details of the provinces were extracted from the "institution" field within the Retraction Watch Database. A total of 14,445 retraction records were ultimately included in our study. The selection process of retracted articles is detailed in Fig 1.

### Statistical analysis

Microsoft Excel 2016 was used for data management and cleaning. Subsequently, a descriptive analysis was conducted using R software (version 4.2.1, R Foundation for Statistical Computing) and Origin pro-2024 b (Origin Lab Co., Northampton, MA, USA), including the number of retracted articles, the number of major research institutes, the reasons for retraction, the subjects, publishers, and journals. The trend surface analysis is a multivariate statistical method based on the least squares approach, and this method was employed to accurately fit the regional distribution and temporal trends in sample data [29]. By considering the number of retracted articles with Chinese authors across 31 provinces along with the corresponding longitude or latitude of each province, we established a binary polynomial regression model and presented it through a three-dimensional visualization to effectively demonstrate the overall spatial trend of retraction occurrences. The gravity center concept was based on the fulcrum that maintains balance and arises from the mechanical field of physics. The gravity center model is an effective analytical method that is used to explore temporal-spatial distribution characteristics; it can accurately and concisely describe the distribution law of elements with a reasonable combination of time and space [30]. Depending on the weight of the number of retracted articles per year in each province, we estimated the center of gravity for retraction each year in China. By drawing the traces described by these nodes, we reconstructed the spatial pattern of displacement for each indicator during 2012–2023.

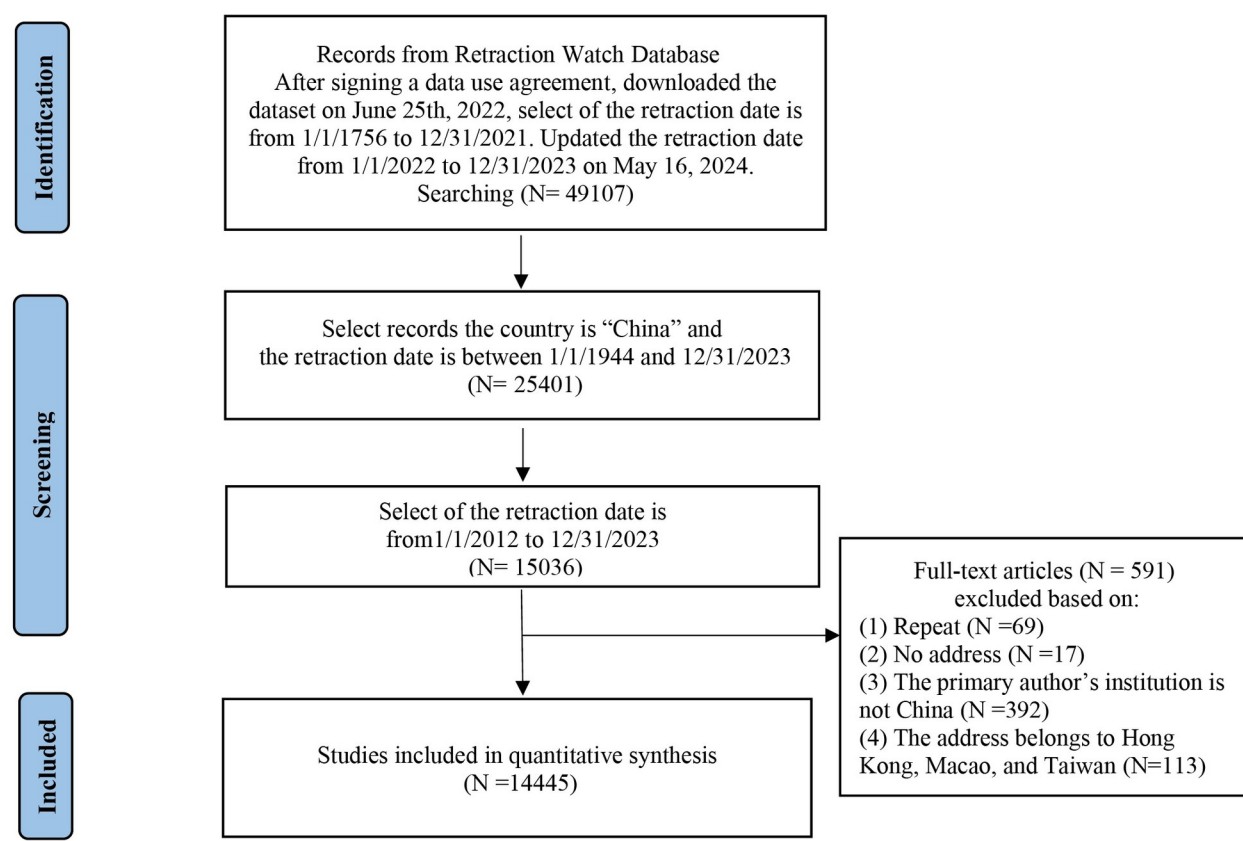

**Fig 1. The selection process of retracted articles from the Retraction Watch Database.**

The spatial autocorrelation analysis of the number of retractions in different provinces across the country was conducted using ArcGIS 10.3 software to unveil the spatial distribution patterns of retraction and to identify areas at risk for retraction [31]. Moran's I was used to evaluate spatial autocorrelation, ranging from -1 to 1. A positive value indicates a positive spatial autocorrelation, while a negative value suggests a negative autocorrelation; a value of zero implies a random spatial distribution with no autocorrelation. Larger absolute values of Moran's I indicate stronger spatial autocorrelation. Statistical significance for Moran's I was set at $P < 0.05$ and Z value >1.96.

Next, the local autocorrelation was analyzed using the Anselin local Moran's I and Getis-Ord Gi* [32]. Anselin local Moran's I was used to determine the locations of the clusters. The high-high (HH), low-low (LL), and outlying local cluster (high-low (HL) and low-high (LH)) locations were visualized with local indicators on a spatial association (LISA) cluster map. Getis-Ord Gi* statistics can measure the aggregation degree of high or low values and detect hot spots and cold spots [33]. A Z-test was conducted for the Gi parameter. Statistically significant hot spots were identified (meaning the provinces had a high number of retracted articles) if the Z value > 1.96, while statistically significant cold spots were identified (meaning the provinces had a low number of retracted articles) if the Z value <-1.96. The corresponding Z values of 90%, 95%, and 99% confidence interval (CI) for the Getis-Ord Gi* were ±1.65, ±1.96, and ±2.58, respectively. The higher the statistically significant positive Z-score, the tighter the clustering of high values (hot spots), and the lower the statistically significant negative Z-score, the tighter the clustering of low values (cold spots).

## Results

### Temporal and spatial trends in the number of retracted articles in China

According to the Retraction Watch Database, from 2012–2023 there were 14,445 retracted articles written by authors from China's 31 provinces. The number of retracted articles remained relatively stable from 2012 to 2018, ranging between 253 and 325, as shown in S1 Fig. However, from 2019 there was a sharp surge in retractions within China's scientific community, with an alarming peak of 5,668 retractions in 2023. Concurrently, based on Web of Science and Scopus, we retrieved the number of publications by China (see S1 Table), there was a substantial growth in the number of scientific articles originating from China during this period, increasing from 417,445 to 1,073,392 according to Scopus. Correlation analysis reveals a significant positive association between the number of retractions and the number of publications during the 2018 to 2023 period (coefficient = 0.903, $P$ = 0.014). From 2012 to 2023, the retraction rate of scientific articles in China was between 0.05% and 0.53%, and the average retraction rate was 0.14%. From 2012 to 2019, the retraction rate was relatively stable, with an average retraction rate of 0.10%. However, the retraction rate rose sharply after 2019 and reached 0.53% in 2023, and the growth rate of retracted articles (91.92%) outpaced the growth rate of published papers (32.3%) during 2019–2023, as shown in Fig 2.

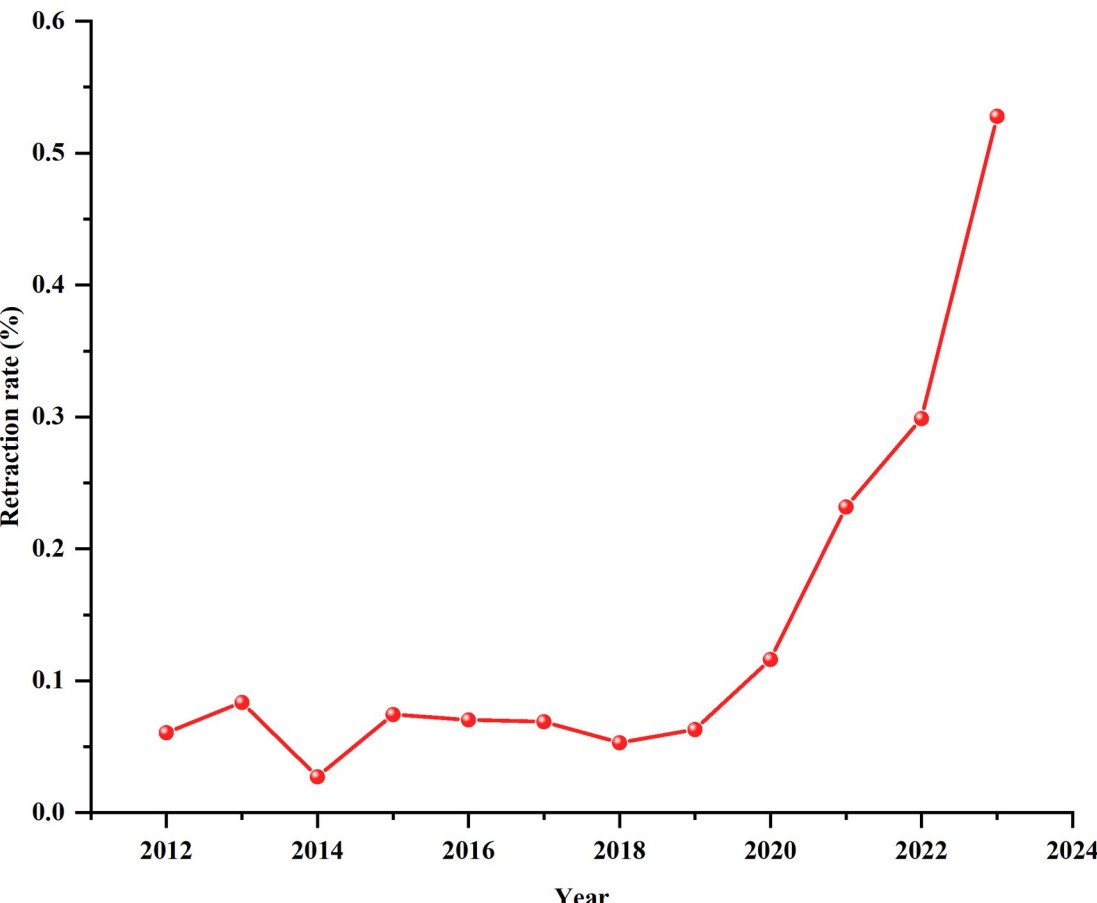

**Fig 2. The trend of retraction rate in China over time.** Note: The number of publications with Chinese authors each year was obtained from Scopus. The retraction rate is the ratio of the number of retracted articles to the total number of publications.

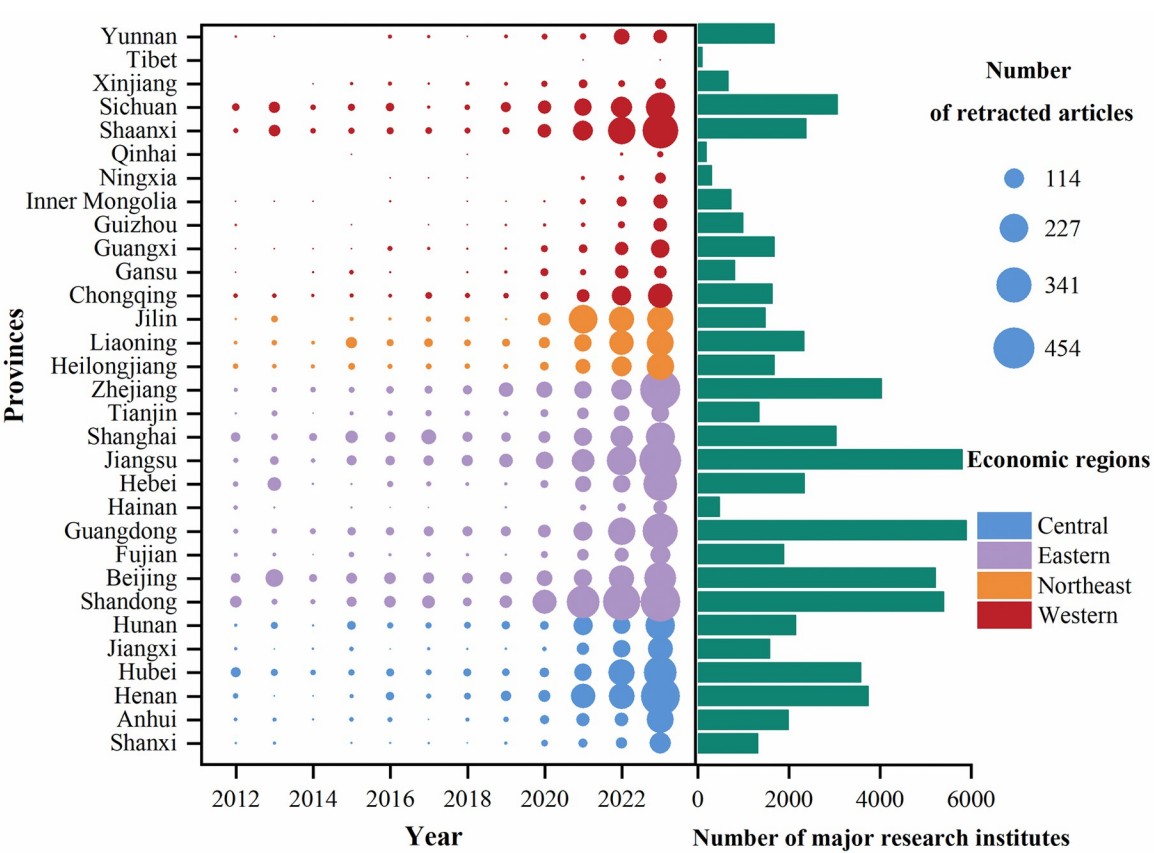

**Fig 3. Spatial distribution of the number of retracted articles and the number of the major research institutes across China's 31 provinces from 2012 to 2023.** Note: According to the definitions provided by National Bureau of Statistics, all provinces were categorized into four regions: the East, the West, the Central and the Northeast. The point of interest (POI) datasets regarding the number of major research institutions, including both research institutes and institutions of higher learning, were obtained from Google Maps in June 2023.

There is an imbalance in the geographical distribution of the number of retracted articles in mainland China. Before 2020, the number of retracted articles from each province in China was fewer than 114. Starting from 2020, the number of retractions in the eastern region gradually increased, and the number of provinces with more than 227 retractions by 2023 increased from one in 2021 to ten provinces, including Jiangsu, Zhejiang, Shandong, Henan, Shaanxi, Guangdong, Hebei, Hunan, Beijing, and Hubei. Additionally, the allocation of major research institutions across China is uneven, with 51.02% located in the eastern region, 20.66% in the central region, 20.44% in the western region, and 7.89% in the Northeast. More than 2000 major scientific research institutions province are mainly concentrated in the eastern provinces, such as Shandong, Beijing, Guangdong, Hebei, Jiangsu, Shanghai, and Zhejiang (see Fig 3).

## Analysis of subjects, publishers, and institutions involved in retracted articles in China

A retracted article may have one or more retraction subjects, each of which is counted separately, for a subject of 41,657. Analysis of the subject of retracted articles from Chinese authors between 2012 and 2023 reveals that retraction was most common in basic life sciences (40.66%), health sciences (21.46%), business and technology (20.32%), with lower percentages

in physical sciences (7.12%), environmental sciences (1.81%), social sciences (7.37%), and humanities (1.25%). The basic life sciences are mainly focused on Cellular (27.38%), Genetics (21.46%), and Cancer (19.39%). As shown in the S2 Table.

A total of 14,445 retracted articles were published by 203 publishers from 2012–2023. The top 10 publishers involved in the retractions are presented: Hindawi (30.90%), Springer (14.57%), Elsevier (9.74%), Taylor and Francis (5.24%), Wiley (4.78%), IOP Publishing (3.99%), Spandidos (3.16%), SAGE Publications (3.07%), IEEE (2.74%), Association for Computing Machinery (ACM) (2.35%). The retractions involved 6466 institutions during 2012–2023, the top 10 institutions with the highest number of retractions include China-Japan Union Hospital of Jilin University, The First Hospital of Jilin University, The First Affiliated Hospital of Zhengzhou University, Central South University, and so on. Please refer to the S3 and S4 Tables for specific details.

## The reasons for retractions

This study employs the categorization of the reasons for retraction found in the Retraction Watch Database, with specific reasons detailed at https://retractionwatch.com/retraction-watch-database-user-guide/retraction-watch-database-user-guide-appendix-b-reasons/. A retracted article may have one or more reasons for retraction, and each retraction reason is counted separately, with a total frequency of 62,496 reasons for retraction. Given the limitations of the Retraction Watch Database's classification of reasons, we have standardized the categorization into nine distinct categories [34]. Please refer to the S5 Table for specific details. The most common reason for retraction was fake data, followed by the other reasons (the top three reasons were concerns/issues about referencing/attributions, paper mill, and randomly generated content), as shown in the S6 and S7 Tables. The proportion of the number of retracted articles with unknown reasons is reduced. The top three reasons for retractions from 2012 to 2014 were plagiarism, fake data, and duplicate publication (excluding other or unknown reasons for retraction), respectively, while the percentage of retractions due to plagiarism decreased. The top three reasons for retractions between 2021 and 2023 (excluding other or unknown reasons for retraction) were fake data, fake review process, and duplicate publication. Fake review process was the most common reason in 2015 and 2017.

## Spatial trend surface analysis

The trend surface fitting results indicate that the spatial trend of 31 provinces in China demonstrates an inverted 'U' type distribution in the north-south direction from 2012 to 2023. Moreover, there is a noteworthy spatial directivity pattern in the east-west direction, with elevated values in the east and lower values in the west. For instance, in 2023, as longitude increases, the number of retractions also increases; meanwhile, as dimension decreases, the number of retractions initially rises and then declines. High-heat areas for the number of retractions are identified in the eastern and central regions. In addition, the trend line alterations vary in differing directions, with a steeper transition of the trend surface in the east-west direction and a comparatively smoother transition in the north-south direction. Therefore, the difference in the number of retractions is more pronounced in the east-west direction, as depicted in Fig 4.

## Gravity center movement

The movement of the number of retracted articles in China from 2012 to 2023 is shown in the S2 Fig The gravity center tracks are located at the junction of Anhui, Shandong, and Fujian, deviating from the geometric center of China (located in Gansu Province). This implies that the number of retracted articles in the eastern region was higher than those in the western

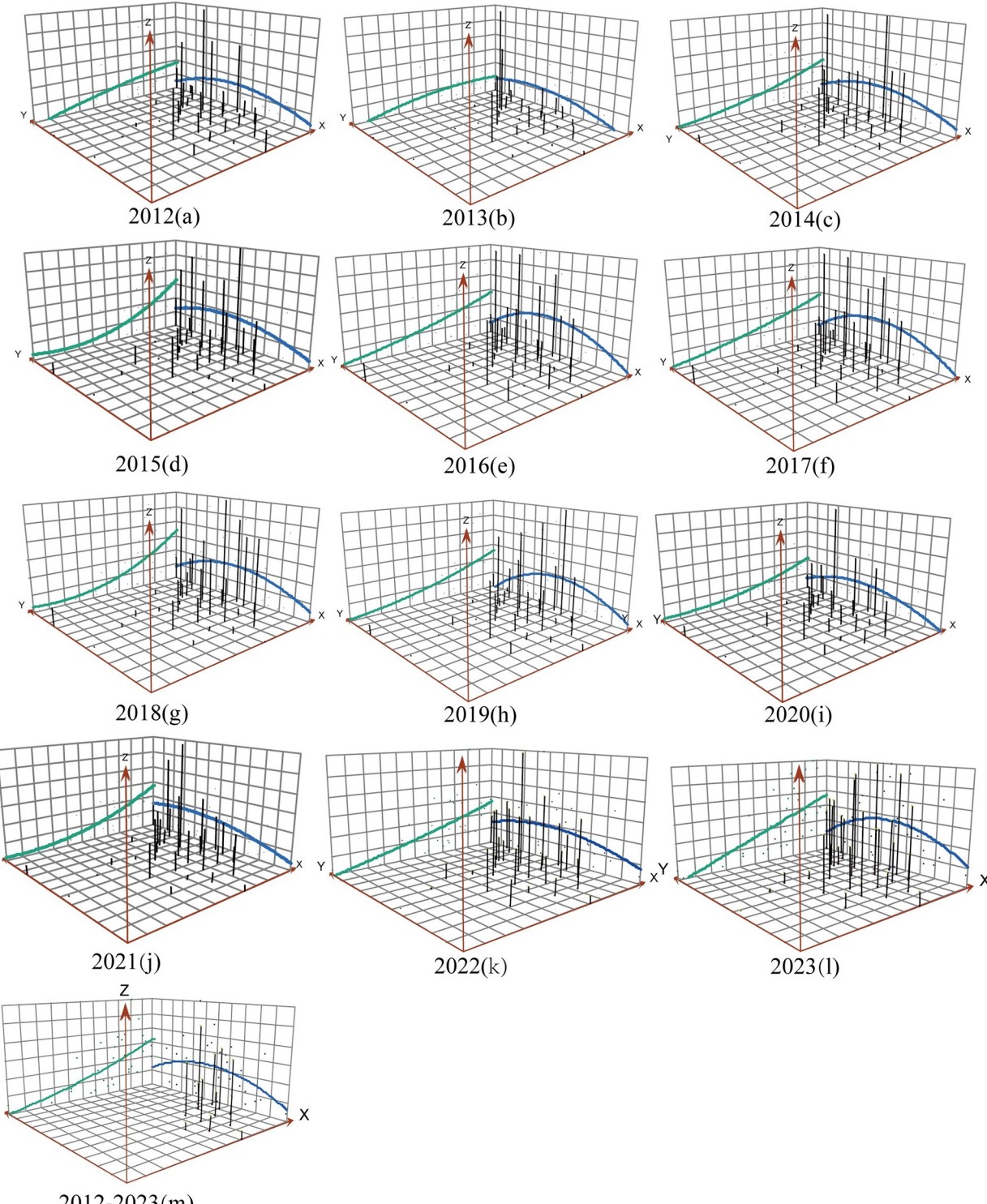

**Fig 4. Spatial trend analysis of retraction numbers across the 31 provinces of China from 2012 to 2023.** Note: The X-axis and Y-axis represent longitude and latitude, and the Z-axis represents the number of retracted articles in each year. The green line represents the east-west direction, and the blue line represents the north-south direction. A point (X, Y, Z) in the three-dimensional space represents the number of retracted articles in a certain research area. Twelve-Year Period: 2012–2023.

region. Regarding the direction of gravity center migration, the gravity center of the number of retracted articles in the longitude direction fluctuated slightly and moved 0.0232˚ eastward, while in the latitude direction, there was less variation with a 0.07969˚ southward movement from 2012–2023.

## Global spatial autocorrelation analysis

The findings of the global autocorrelation analysis indicate that the annual Moran's I was statistically significant in the years 2017, 2018, 2019, 2021, and 2023, and in the period spanning 2012–2023 (Z>1.96, P<0.05). The Moran's I values ranged between 0.192462 and 0.680713. The spatial distribution exhibited a positive correlation, suggesting that the number of retractions among the 31 provinces of China was nonrandomly distributed, with notable regional aggregation characteristics in recent years. However, in the years 2012, 2013, 2014, 2015, 2016, 2020, and 2022, there were no statistically significant differences (Z<1.96, P>0.05) (see Table 1).

## Local spatial autocorrelation analysis

To identify the specific areas and types of spatial clustering local spatial autocorrelation analysis was conducted. As shown in Table 2, the findings of the local spatial autocorrelation analysis indicate that the high-high (H-H) cluster was initially located in Jiangsu in 2017, and expanded to include areas around Jiangsu and Shanghai in 2018 and 2019. However, in 2021, no high-high cluster areas were observed; the H-H cluster areas expanded to Henan, Jiangsu, Shandong, Anhui, and Shanghai in 2023. Overall, the H-H cluster areas in 2012–2023 were found in Jiangsu and Shandong, which suggests that the high proportions of retracted articles were clustered between adjacent provinces (positive correlation). On the other hand, the low-low (L-L) cluster areas were found in various regions over the years, including Xinjiang Uygur Autonomous Region, Tibet Autonomous Region, Sichuan, Yunnan, Gansu, and Qinghai in 2017. In 2018, the L-L cluster areas were observed in the Xinjiang Uygur Autonomous Region, Gansu, Sichuan, and Yunnan. In 2019, L-L cluster areas were located in the Xinjiang Uygur

**Table 1. Global spatial autocorrelation analysis of the retraction count across 31 provinces in China.**

| Year | Moran's I | Z | P |
|---|---|---|---|
| 2012 | 0.032254 | 0.591990 | 0.553857 |
| 2013 | 0.140519 | 1.708202 | 0.087599 |
| 2014 | -0.001321 | 0.280117 | 0.779388 |
| 2015 | 0.071429 | 0.906566 | 0.364636 |
| 2016 | 0.187304 | 1.902974 | 0.057044 |
| 2017 | 0.241687 | 2.484334 | 0.012979 |
| 2018 | 0.267516 | 2.583137 | 0.009791 |
| 2019 | 0.680713 | 6.092060 | <0.000001 |
| 2020 | 0.151326 | 1.772237 | 0.076355 |
| 2021 | 0.192462 | 2.043047 | 0.041048 |
| 2022 | 0.164358 | 1.740579 | 0.081757 |
| 2023 | 0.347442 | 3.218627 | 0.001288 |
| 2012–2023 | 0.279253 | 2.679563 | 0.007372 |

Note: The global autocorrelation analysis conducted in 2012, 2013, 2014, 2015, 2016, 2020and 2022 shows that Z scores are less than 1.96 and P values are greater than 0.05, indicating no evidence of local autocorrelation and hot spot analysis. Twelve-Year Period: 2012–2023.

**Table 2. Local spatial autocorrelation analysis of the retraction counts in the 31 provinces of China.**

| Year | Type of Clustering | | | |
|------|------------------|---|---|---|
| | **H-H clustering** | **H-L clustering** | **L-H clustering** | **L-L clustering** |
| 2017 | Jiangsu | Shaanxi | | Xinjiang Uygur Autonomous Region, Tibet Autonomous Region, Sichuan, Yunnan, Gansu, and Qinghai |
| 2018 | Jiangsu, Shanghai | | Anhui | Xinjiang Uygur Autonomous Region, Gansu, Sichuan, and Yunnan |
| 2019 | Jiangsu, Shanghai | Sichuan, Liaoning | Anhui | Xinjiang Uygur Autonomous Region and Inner Mongolia Autonomous Region |
| 2021 | | Sichuan | Anhui | Xinjiang Uygur Autonomous, Qinghai, Gansu, and Yunnan |
| 2023 | Henan, Jiangsu, Shandong, Anhui, and Shanghai | Sichuan | | Xinjiang Uygur Autonomous Region, Qinghai, Tibet Autonomous Region, and Gansu |
| 2012–2023 | Jiangsu, Shandong | Sichuan | Anhui | Xinjiang Uygur Autonomous, Qinghai, and Gansu |

Autonomous Region and Inner Mongolia Autonomous Region. L-L cluster areas were predominantly found in Xinjiang Uygur Autonomous, Qinghai, Gansu, and Yunnan in 2021, while they were located in Xinjiang Uygur Autonomous Region, Qinghai, Tibet Autonomous Region, and Gansu in 2023. Over the years 2012–2023, the L-L cluster areas were Xinjiang Uygur Autonomous, Qinghai, and Gansu, illustrating how the low proportions of retracted articles were clustered among adjacent provinces (positive correlation). The high-low (H-L) cluster areas, in Shaanxi in 2017, and Sichuan and Liaoning in 2019, and primarily in Sichuan in 2021, 2023, and 2012–2023, emphasized that the provinces with high numbers of retracted articles were surrounded by provinces with low numbers of retracted articles (negative correlation). Lastly, the low-high (L-H) cluster areas, which were observed in 2018, 2019, 2021, and 2012–2023, were primarily located in the province of Anhui. This L-H clustering area suggests that provinces with low numbers of retracted articles are surrounded by provinces with high numbers of retracted articles (negative correlation).

## Getis-Ord Gi* analysis

The Getis-Ord Gi* statistical analysis suggests that the regions with the highest number of retracted articles are concentrated in hotspots, including Jiangsu and Shanghai in 2017, Jiangsu, Shanghai, and Anhui in 2018, Shandong, Jiangsu, Anhui, and Shanghai in 2019, and Shandong, Jiangsu, Anhui, and Henan in 2021. In 2023 these hotspots expanded to include Shandong, Jiangsu, Anhui, Henan, and Shanghai. Overall, the hotspot areas for 2012–2023 included Shandong, Jiangsu, Shanghai, Henan, and Anhui. In contrast, the regions exhibiting low values of retracted articles are referred to as cold spot areas. There were no cold spot regions in 2017, 2018, 2019, and 2021, but in 2023, Xinjiang Uygur Autonomous Region, Tibet Autonomous Region, and Qinghai became cold spot regions. The cold spot for 2012–2023 includes Xinjiang Uygur Autonomous Region and Qinghai. The details are shown in Table 3.

## Discussion

In this study, we examined the spatial distribution characteristics of the number of retracted articles by Chinese authors in 31 provinces in China from 2012 to 2023 using ArcGIS10.3. The results show that the number of retracted articles by authors in China's 31 provinces gradually increased from west to east from 2012 to 2023. Clustering areas with high numbers of retracted articles were mainly in Shandong, Jiangsu, Anhui, Henan, and Shanghai, and the research articles were particularly within the subject area of basic life sciences. Focusing on these highly aggregated areas is crucial for implementing precise prevention and control measures.

**Table 3. Getis-Ord Gi \* analysis results of retracted articles from 31 provinces in China.**

| Type of Clustering | Year | | | | | |
|---|---|---|---|---|---|---|
| | **2017** | **2018** | **2019** | **2021** | **2023** | **2012–2023** |
| Hot spot 99% CI | Jiangsu, Shanghai | Jiangsu, Shanghai | Jiangsu, Shanghai, Anhui | Shandong | Shandong, Jiangsu, Anhui | Shandong, Jiangsu, Anhui |
| Hot spot 95% CI | | Anhui | Shandong | Jiangsu, Anhui, Henan | Henan, Shanghai | Henan, Shanghai |
| Hot spot 90% CI | | Zhejiang | Zhejiang | | | |
| Cold spot 99% CI | | | | | Xinjiang Uygur Autonomous Region | |
| Cold spot 95% CI | | | | | Tibet Autonomous Region, Qinghai | Xinjiang Uygur Autonomous Region, Qinghai |
| Cold spot 90% CI | Xinjiang Uygur Autonomous Region, Tibet Autonomous Region, Sichuan, Gansu, Qinghai | Xinjiang Uygur Autonomous Region, Gansu, Sichuan, Yunnan | Inner Mongolia Autonomous Region | Xinjiang Uygur Autonomous Region, Tibet Autonomous Region, Sichuan, Qinghai | Gansu, Sichuan, Yunnan | Tibet Autonomous Region, Gansu, Sichuan, Yunnan |

The present study found that the number of retracted articles remained relatively stable from 2012 to 2018 at between 253 and 325. Nevertheless, beginning in 2019, there was a sharp surge, with an alarming peak of 5,668 retracted articles in 2023. Our results are consistent with those from previous studies [35]. The factors contributing to the escalation are multifactorial. Primarily, the number of retracted articles showed a substantial correlation with the volume of articles published [36]. In our study, we found that the number of retracted articles was significantly correlated with the volume of articles published in the years 2018–2023. The number of Chinese publications reached 1,073,392 in Scopus in 2023. Furthermore, since 1990, China has implemented a heightened level of research funding and incentives for researchers [37]. The increase in the number of retracted articles may also be a consequence of the "publish or perish" culture, rewarding quantity over quality in academic productivity to attain graduation, promotion, and rewards, which may lead to academic misconduct and eventually to an increased frequency of retractions [38]. Innovation in monitoring and reporting research misconduct, including extensive testing of Retraction Warch Databases [39], as well as increasing scrutiny from China and publishers (such as Hindawi) [4, 16], and the widespread use of plagiarism detection software [40, 41], may contribute to the detection and reporting of a greater number of retracted articles. There is one other point worth noting our study showed that the retraction rate rose sharply after 2019 and reached 0.53% in 2023. Above we have analyzed why the peak in retractions in 2023 is associated with increased research output, "publish or perish" culture, and advancements in detecting and addressing research misconduct. In addition, the growth rate of retracted articles outpaced the growth rate of published papers during 2019–2023 compared to 2012–2018. These may be the reasons for a sharp increase in the retraction rate.

In addition, this study found that a fake review process was the most common reason for retraction in 2015 and 2017 and one of the top three reasons between 2021 and 2023. These findings are similar to Qi et al, who found that the percentage of retracted articles due to falsified peer review peaked in 2015 [42, 43]. During 2015–2017, a significant number of retractions due to peer evaluation occurred; for example, in March 2015, Bio Med Central retracted 43 articles, 41 of which were from Chinese authors. With media exposure of such retractions, more journal editors have recognized the flaws in the peer review system and their negative effects. China attaches great importance to scientific research integrity and has developed a

series of relevant policies. In December 2015, seven Chinese governmental agencies coalesced to propose five categories of publishing conduct that must be unequivocally prohibited, including fake peer review [44]. In 2018, legislation targeting scientific fraud was reinforced by introducing penalties that extend beyond the academic and professional realms [45]. Our results show that the proportion of retracted articles because of fake peer review decreased between 2018 and 2020. With the increase in the number of retracted articles in China, as well as the strict scrutiny of retractions and research misconduct in China [16], the proportion of peer review may have increased in recent years.

Paper mill services appear to include data fabrication, authorship for sale, fake peer review, and citation schemes. The articles often resemble each other in multiple ways, they tend to pass through peer review smoothly, and the published articles are generally well cited. is that its share will increase in 2021 [46, 47]. Our study showed that paper mills were one of the main reasons for retraction and the proportion of retractions attributable to paper mills has been progressively increasing over time. Previous studies have shown that more than 400,000 studies with textual similarities to paper mill articles have been published in the past 20 years, with 70,000 published in 2022 alone [48]. Paper mill retractions mostly come from Chinese organizations [49]. Moreover, our study showed that the basic life sciences field continues to have a high frequency of retractions, this is similar to the research findings of Yang Yao et al [50]. Previous studies have also found that biomedical articles are one of the hot topics in China [51]. With use of paper mills being an important reason for retraction in the basic life sciences [52]. Moreover, retracted articles in recent years are related to the field of basic life sciences. However, most authors of identified paper mill articles are hospital affiliates [49, 53], and many of such articles have been published in basic scientific journals such as *Cell Molecular Biology* or *Biochemistry*. Paper mill customers—students and scientists—are pressured to publish in SCI publications by their academic or government institutions or university-affiliated hospitals [54], Therefore, this issue should be given full attention in the future.

Our findings indicate a higher number of retracted articles in eastern China compared to that observed in western China, with trend analyses showing that this gradually diminished from east to west. Possible reasons for this trend are as follows. On the one hand, the data reveals a consistent growth in national R&D personnel and financial expenditure on science and technology, albeit with a distinct pattern characterized by higher funding in the east and comparatively lower funding in the west [55]. On the other hand, the regional distribution of science and technology platforms has apparent gradient differentiation. The first echelon, including Beijing, Shanghai, and Jiangsu, accounts for 45.8% of the total [56]. The local autocorrelation analysis for the period 2012–2023 shows that the high-value clusters are in Jiangsu and Shandong, while the e low-value clusters are in Tibet Autonomous Region, Qinghai, and Gansu. The Getis-Ord Gi* analyses show that Shandong, Jiangsu, Anhui, Henan, and Shanghai belong to the hotspot area, while Xinjiang Uygur Autonomous and Qinghai belong to the cold spot area in our study. Compared to other regions, Jiangsu, Shanghai, and Shandong are hotspots for article retractions. These provinces are characterized by a higher concentration of scientific research institutions, greater availability of research funding, and an influx of scientific and technological talents. The intense competition and high pressure associated with scientific research may lead to undesirable phenomena. In the future, it is essential to further analyze the causes of retractions in these highly aggregated areas taking into account the specific context of each province, to implement more targeted prevention and control measures.

Therefore, actively implementing measures to reduce the emphasis on publications in professional title evaluations and other utilitarian requirements may significantly decrease the incidence of retractions. Furthermore, enhancing the rigorous review and verification of

article data prior to publication could serve as a strong deterrent against data fabrication and related issues. Additionally, it is essential to strengthen punitive measures and promote education on research integrity.

This study has significant limitations. The format of retraction notices varies across journals and may not include all information about the original publications. Therefore, although the Retraction Watch Database is continuously updated and verified, some retraction notices may be inadvertently omitted. Furthermore, the lack of accessible data on the number of publications by Chinese authors in each province makes it challenging to calculate the relative rate of retractions, which could introduce bias. The primary innovation of this study is the use of spatial statistical analysis techniques to examine the spatial distribution of retracted articles across 31 provinces and identify hotspot areas that could inform timely adjustments or enhancements to prevention and control measures. However, it is essential to note that the change in the number of retracted articles is a complex issue that warrants a holistic and comprehensive analysis, taking into account various factors.

## Conclusion

This study mainly attempts to remedy the current lack of research about the spatial distribution of retracted articles by Chinese authors. Notably, there is a discernible spatial clustering among these retractions, with a gradual increase in the number of retracted articles from west to east. Shandong, Jiangsu, Shanghai, Henan, and An-hui are the hotspots for retractions. In view of this, the causes of retraction should be further analyzed for these highly aggregated areas within the context of each province in order to take more precise prevention and control measures.

## Supporting information

**S1 Table. The number of publications (Retraction rate) by Chinese authors from 2012–2023 based on Scopus and Web of Science.**
(DOCX)

**S2 Table. The distribution subject of retracted articles from Chinese authors between 2012 and 2023(N/%).**
(DOCX)

**S3 Table. The top 10 publishers involved retracted articles from Chinese authors published between 2012 and 2023.**
(DOCX)

**S4 Table. The top 10 institutions involved retracted articles from Chinese first authors institutions between 2012 and 2023.**
(DOCX)

**S5 Table. Standardized reason categorization corresponding to categorization employed by the Retraction Watch Database.**
(DOCX)

**S6 Table. The reasons for retractions.**
(DOCX)

**S7 Table. The top 3 classification of other reasons for retraction.**
(DOCX)

**S1 Fig. The number of publications with Chinese authors and retracted articles from 2012 to 2023.** Note: The number of publications with Chinese authors is based on data from Scopus, retrieved August 29, 2024.
(TIF)

**S2 Fig. The migration tracks the number of retracted articles in China from 2012–2023.**
(TIF)

## Author Contributions

**Conceptualization:** Jiangping Li, Shulan He.

**Data curation:** Liping Shi, Xue Zhang, Xiaojun Ma, Xian Sun.

**Methodology:** Jiangping Li.

**Resources:** Shulan He.

**Supervision:** Jiangping Li.

**Writing – original draft:** Liping Shi.

**Writing – review & editing:** Liping Shi, Shulan He.

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
