## [Decision Letter · Decision Letter 0]

9 Aug 2024

PONE-D-24-27000Mapping retracted articles and exploring regional differences in China, 2012-2023PLOS ONE

Dear Dr. He,

Thank you for submitting your manuscript to PLOS ONE. After careful consideration, we feel that it has merit but does not fully meet PLOS ONE’s publication criteria as it currently stands. Therefore, we invite you to submit a revised version of the manuscript that addresses the points raised during the review process.

We look forward to receiving your revised manuscript.

Kind regards,

Robin Haunschild

Academic Editor

PLOS ONE

Journal Requirements:

Reviewers' comments:

Reviewer's Responses to Questions

**Comments to the Author**

1. Is the manuscript technically sound, and do the data support the conclusions?

Reviewer #1: Yes

Reviewer #2: Yes

Reviewer #3: Yes

2. Has the statistical analysis been performed appropriately and rigorously? 

Reviewer #1: Yes

Reviewer #2: Yes

Reviewer #3: Yes

3. Have the authors made all data underlying the findings in their manuscript fully available?

Reviewer #1: Yes

Reviewer #2: Yes

Reviewer #3: Yes

4. Is the manuscript presented in an intelligible fashion and written in standard English?

Reviewer #1: Yes

Reviewer #2: Yes

Reviewer #3: Yes

5. Review Comments to the Author

**Reviewer #1: **Mapping retracted articles and exploring regional differences in China, 2012-2023

I recently read the article “Mapping retracted articles and exploring regional differences in China, 2012-2023”. I understand that this is a pressing topic, and I therefore tried to provide my review quickly. Overall, I liked reading the paper; it made a meaningful contribution.

Introduction

• (pg. 9, ln. 57) cite more literature. In many studies China tops the list of retraction.

Methodology

• (pg. 11, ln. 87) provide retraction watch database website link

• (pg. 11, ln. 97) cite studies conducted using retraction watch database

Statistical Analysis

• (pg. 13, ln. 138) cites literature that used similar tests to substantiate the analysis.

Results

• (pg. 14, ln. 153) what could be the reason(s) for a sharp surge in retractions within 154 China’s scientific community, with an alarming peak of 5,668 retractions in 2023? Justify.

• (pg. 15, ln. 161) “the retraction rate rose 162 sharply after 2019 and reached 0.64% in 2023” reason(s) for a sharp increase in the retraction rate?

Conclusion

• Suggesting that authors incorporate broader reflection on the implications of your findings for the field as a whole is needed and the concluding sentences should come up with some reflections regarding your takeaways from the study.

• Although the subject of the study is worthy of further exploration, the current paper does not offer new insights that address research gaps or provide practical implications for policymakers, institutions and academicians.

Comment

• There is inconsistency in in-text citation. For example

• (pg. 9, ln 42) is cited as “an aspect of science(5)”

• (pg. 16, ln 192) is cited as “distinct categories (24)”

• (pg. 9, ln 44) is cited as “eliminate the scientific environment(6, 7)”

• (pg. 9, ln 57) is cited as “retracted publications(13) (14)”

• Suggesting author(s) to check and follow the authorship guidelines.

I appreciate the author’s consideration of these comments, as they are only intended to strengthen the manuscript.

**Reviewer #2:** I appreciate the authors for conducting a study of retracted articles written by Chinese authors using data from the Retraction Watch database. This study focuses on retracted articles written by Chinese authors from 31 provinces in 25 mainland China during the study period between January 1, 2012, and December 31, 2023. The author may consider explaining the following points:

The author conducted a subject-wise analysis of retracted articles and the reasons behind them, focusing on Chinese authors from 31 provinces over a 12-year period. Does this analysis alone suffice to justify the significance of retracted articles and the exploration of regional differences in China?

It lacks institutes, publishers, and authors in particular; it should have at the very least list the top 10 of these.

The authors used the words “retracted article” and “retracted manuscript” in most places. For example, on page 9, line 179, the author writes about the subject analysis of retracted manuscripts in China, and on page 9, line 180, the author writes about the subject analysis of retracted articles from Chinese authors. The research is based on retracted articles, not manuscripts.

Figure 4: Autocorrelation analysis mapping is difficult to understand; you can provide a table or provide more explanations.

Supplement Information Fig. S1, Page 7, Line 21 and 22: Authors has mentioned data source of the research is from Retraction Watch database for retracted articles and the number of publications with Chinese authors is based on data from the Web of Science. Scopus and Web of Science are the two major databases, Scopus covers more peer-reviewed journals than Web of Science, my question is why authors chose Web of Science for year wise publication.

Supplement Information Fig. S1, Page 7, Lines 21 and 22: The authors have mentioned that the data source for the research is from Retraction Watch database for retracted articles, and the number of publications with Chinese authors is based on data from the Web of Science. Scopus and Web of Science are the two major databases. Scopus covers more peer-reviewed journals than Web of Science. My question is why authors chose Web of Science for year-wise publication.

Overall, I suggest the author may include additional up-to-date references on retracted articles.

**Reviewer #3:** The manuscript dealt with retracted articles authored by Chines authors and the topic is of interest and I recommend to publish it. It needs minor revision before publication.

1. In page 9, a sentence reads "No province had more than 106 retractions before 2020" which is not clear.

2. In page 9, there is note below figure 2, "Note: The number of publications with Chinese authors each year was obtained from the Web of Science" which is contradictory to the data source mentioned in Methodology section.

3. Subject analysis needs more details.

4. In page 10, an in-text citation "distinct categories24" which is not formatted and the provided reference is not appropriate. Please check the reference provided.

6. PLOS authors have the option to publish the peer review history of their article (what does this mean?). If published, this will include your full peer review and any attached files.

Reviewer #1: **Yes: **Dr. Somipam R Shimray

Reviewer #2: **Yes: **Dr. SIVA N

Reviewer #3: No

---

## [Author Response · Author response to Decision Letter 0]

19 Sep 2024

Response to Reviewers

Dear Editor,

We are grateful for the valuable comments and suggestions given by the reviewers concerning our manuscript entitled “Mapping retracted articles and exploring regional differences in China, 2012-2023” (ID: PONE-D-24-27000). We give our responses to the reviewer in this let-ter. We hope that after our replies, our manuscript will be considered for publication in PLOS ONE. The detailed responses are listed below, and revised parts of the manuscript are marked with red color.

Question 1. Figures 3, 5, and 6 in your submission contain [map/satellite] images which may be copyrighted

Response: Thanks. We are sorry that we are unable to obtain permission from the original copyright holder to publish Figures 3, 5, and 6 under the CC BY 4.0 license. We have learned that the map images we utilized must be submitted for review to the com-petent geospatial infor-mation administration department, which holds the authority for map review" (https://www.gov.cn/zhengce/2015-12/14/content_5023591.htm). Moreover, map review applications can only be submitted by legal entities, not by individuals, thus we have revised these figures as follows:

Fig 3 has been replaced with a bubble map plotted using Origin pro 2024b (Origin Lab Co., Northampton, MA, USA, Serial number DL2W8-6089-7609063).

Fig 5 and Fig 6 have beeb replaced with Table 2 and Table 3 to present the results of the Local spatial autocorrelation analysis and Getis-Ord Gi * analysis.

We have checked copyright information on all replacement figures and updated the figure caption with source information 

Question 2. a) Please provide search terms that can be used on the Retraction Watch Database to access the original dataset(s).

b.) Please clarify if you have permission from the original database to reupload their data to Figshare.

Response: Thanks. 

After signing a data use agreement, we downloaded the dataset from the Retrac-tion Watch Database on July 25, 2022 (https://api.labs.crossref.org/data/retractionwatch?name@email.org (add your ‘mail-to’)) and updated the data for 2022 and 2023 on May 16, 2024. The usage agreement for the Retraction Watch Database is shown in Fig S3. The Retraction Watch Data-base was completely open and freely available on September 12, 2023. 

The records downloaded from Retraction Watch Database were filtered by the country “China” and included only those with a retraction date between January 1, 2012, and December 31, 2023. This study focuses solely on the institutions of the first authors of the retracted articles from the 31 provinces in mainland China. The details of the provinces were extracted from the “institution” field within the Retraction Watch Database. A total of 14,445 retraction records were ultimately included in our study. The selection process of retracted articles is detailed in Fig 1.

Additionally, we have reuploaded this data to Figshare. https://doi.org/10.6084/m9.figshare.27044899.

Question 3: Please upload the minimal data set necessary to replicate your study's findings to a stable, public repository (such as Figshare or Dryad) and pro-vide us with the relevant URLs, DOIs, or accession numbers that may be used to access these data.

Response: Thanks. We have uploaded the necessary minimum dataset to Figshare (https:// /doi.org/10.6084/m9.figshare.27044899).

Question 4: Please include captions for your Supporting Information files at the end of your manuscript, and update any in-text citations to match accordingly.

Response: Thanks. We have added captions for the Supporting Information files at the end of the revised manuscript and updated any in-text citations to match ac-cordingly in our revised manuscript.

Reviewer #1:

Question 1: (pg. 9, ln. 57) cite more literature. In many studies, China tops the list of retractions.

Response: Thanks for your suggestion. We have cited more literature in the re-vised manuscript to support the statement that China tops the list of retractions 

The literatures were cited as follows:

Van Noorden R. More than 10,000 research papers were retracted in 2023-a new record. Nature 2023;624(7992):479-481.https://doi.org/10.1038/d41586-023-03974-81.

Zhang Q, Abraham J, Fu H. Collaboration and its influence on retraction based on retracted publications during 1978–2017. Scientometrics 2020; 125:213-232. https://doi.org/10.1007/s11192-020-03636-w.

Ataie-Ashtiani B. Chinese and Iranian Scientific Publications: Fast Growth and Poor Ethics. Sci Eng Ethics 2017;23(1):317-319. https://doi.org/10.1007/s11948-016-9766-1.

Xiao Y, Chen J, Wu XH, Qiu QM. High retraction rate of Chinese articles: it is time to do something about academic misconduct. Postgrad Med J 2022;98(1163):653-654. https://doi:10.1136/postgradmedj-2021-140853. PMID: 37062972.

Eldakar MAM, Shehata AMK. A bibliometric study of article retractions in technology fields in developing economies countries. Scientometrics 2023;128(11):6047-6083. https://doi.org/10.1007/s11192-023-04823-1.

Question 2: (pg. 11, ln. 87) Provide retraction watch database website link

Response: Thanks for your suggestion. We have added the Retraction Watch Database website link on page 5, line 93 in the revised manuscripts, as follows:

We utilized the Retraction Watch Database for our study (http://www.retractiondatabase.org/RetractionSearch.aspx).

Question 3: (pg. 11, ln. 97) cite studies conducted using retraction watch data-base 

Response: Thanks for your suggestion. We have cited studies conducted using the Retraction Watch Database in the revised manuscript, as follows:

Yang W, Sun N, Song H. Analysis of the retraction papers in oncology field from Chinese scholars from 2013 to 2022. J Cancer Res Ther 2024;20(2):592-598. https://doi:10.4103/jcrt.jcrt_1627_23. Epub 2024 Apr 30. PMID: 38687929.

Kocyigit BF, Zhaksylyk A, Akyol A, Yessirkepov M. Characteristics of Retracted Publications From Kazakhstan: An Analysis Using the Retraction Watch Database. J Korean Med Sci 2023;38(46): e390. https://doi:10.3346/jkms.2023.38. e390. PMID: 38013646; PMCID: PMC10681843.

Zhong QY, Zhang XY, Luo HH, Jiang X, Zeng XY, Jiang J, et al. [Analysis of the characteristics of retracted scientific papers in the field of global liver diseases pub-lished by Chinese scholars]. Zhonghua Gan Zang Bing Za Zhi 2023;31(1):96-100. https://doi:10.3760/cma.j.cn501113-20210324-00138. PMID: 36948856.

.

Question 4: (pg. 13, ln. 138) cites literature that used similar tests to substanti-ate the analysis.

Response: Thanks for your suggestion. We have cited literature that used similar tests to substantiate the analysis in the revised manuscripts as follows:

Sako S, Gilano G, Dileba T, Ayenew T, Addis Y. Spatial distribution and deter-minants of exclusive breastfeeding practice among mothers of children under 24 months of age in Ethiopia: spatial and multi-level analysis. BMC Pregnancy Child-birth 2024;24(1):554.

Question 5: (pg. 14, ln. 153) What could be the reason(s) for a sharp surge in retractions within China’s scientific community, with an alarming peak of 5,668 retractions in 2023? Justify.

Response: Thanks for your suggestion. The sharp increase in retractions within China's scientific community, which reached an alarming peak of 5,668 retractions in 2023, may be attributed to multiple factors and we have supplemented in the section of discussion in our revised manuscript. The factors are as follows:

Primarily, the number of retracted articles showed a substantial correlation with the volume of articles published(36). In our study, we found that the number of retracted articles was significantly correlated with the volume of articles published in the years 2018–2023. Notably, the number of Chinese publications reached 1,073,392 in Scopus in 2023. Furthermore, since 1990, China has implemented a heightened level of research funding and incentives for researchers(37). The increase in the number of retracted articles may also be a consequence of the “publish or per-ish” culture, rewarding quantity over quality in academic productivity to attain graduation, promotion, and rewards, which may lead to academic misconduct and eventually to an increased frequency of retractions (38). Innovation in monitoring and reporting research misconduct, including extensive testing of Retraction Warch Databases(39), as well as increasing scrutiny from China and publishers(such as Hindawi) (4, 16), and the widespread use of plagiarism detection software(40-41), may contribute to the detection and reporting of a greater number of retracted arti-cles.

Question 6: (pg. 15, ln. 161) “The retraction rate rose sharply after 2019 and reached 0.64% in 2023” Reason (s) for a sharp increase in the retraction rate?

Response: Thanks for your suggestion. The sharp increase in the retraction rate can be attributed to several key factors, which we have supplemented in the section of discussion in our revised manuscript. As follows:

There is one other point worth noting our study showed that the retraction rate rose sharply after 2019 and reached 0.53% in 2023. Above we have analyzed why the peak in retractions in 2023 is associated with increased research output, “publish or perish” culture, and advancements in detecting and addressing research misconduct. In addition, the growth rate of retracted articles outpaced the growth rate of pub-lished papers during 2019–2023 compared to 2012-2018. These may be the reasons for a sharp increase in the retraction rate.

Question 7: Suggesting that authors incorporate broader reflection on the im-plications of your findings for the field as a whole is needed and the concluding sentences should come up with some reflections regarding your takeaways from the study.

Response: Thanks for your suggestion. We have incorporated broader reflection on the implication of our findings for the field as a whole in the section of discussion in the revised manuscript and modified the conclusion as follows:

This study mainly attempts to remedy the current lack of research about the spa-tial distribution of retracted articles by Chinese authors. Notably, there is a discerni-ble spatial clustering among these retractions, with a gradual increase in the number of retracted articles from west to east. Shandong, Jiangsu, Shanghai, Henan, and An-hui are the hotspots for retractions. In view of this, the causes of retraction should be further analyzed for these highly aggregated areas within the context of each prov-ince in order to take more precise prevention and control measures.

Question 8: Although the subject of the study is worthy of further exploration, the current paper does not offer new insights that address research gaps or provide practical implications for policymakers, institutions, and academicians.

Response: Thanks for your suggestion. We have revised the section of the dis-cussion in the Revised Manuscript to provide practical implications for policymakers, institutions, and academicians. As following:

Therefore, actively implementing measures to reduce the emphasis on publica-tions in professional title evaluations and other utilitarian requirements may signifi-cantly decrease the incidence of retractions. Furthermore, enhancing the rigorous review and verification of article data prior to publication could serve as a strong deterrent against data fabrication and related issues. Additionally, it is essential to strengthen punitive measures and promote education on research integrity.

Question 9: There is inconsistency in in-text citations. For example (pg. 9, ln 42) is cited as “an aspect of science (5)”; (pg. 16, ln 192) is cited as “distinct cate-gories (24)” (pg. 9, ln 44) is cited as “eliminate the scientific environment (6, 7)” (pg. 9, ln 57) is cited as “retracted publications (13) (14)” Suggesting author(s) check and follow the authorship guidelines.

Response: Thanks. We have thoroughly reviewed the manuscript to ensure that all in-text citations are consistent and correctly formatted according to the author-ship guidelines. 

Reviewer #2

Question 1: The author conducted a subject-wise analysis of retracted articles and the reasons behind them, focusing on Chinese authors from 31 provinces over 12 years. Does this analysis alone suffice to justify the significance of retracted ar-ticles and the exploration of regional differences in China? It lacks institutes, pub-lishers, and authors in particular; it should have at the very least listed the top 10 of these.

Response: Thanks for your reasonable proposal. We have added an analysis of institutes, publishers, and authors in particular and listed the top 10 of these in the revised manuscript as follows:

A total of 14,445 retracted articles were published by 203 publishers from 2012 to 2023. The top 10 publishers involved in the retractions are presented: Hindawi (30.90%), Springer(14.57%), Elsevier(9.74%), Taylor and Francis(5.24%), Wiley(4.78%), IOP Publishing(3.99%), Spandidos(3.16%), SAGE Publica-tions(3.07%), IEEE (2.74%), Association for Computing Machinery (ACM) (2.35%). The retractions involved 6466 journals during 2012 -2023, the top 10 journals with the highest number of retractions include China-Japan Union Hospital of Jilin Uni-versity, The First Hospital of Jilin University, The First Affiliated Hospital of Zheng-zhou University, Central South University, and so on. Please refer to the S3 and S4 Tables for specific details.

Question 3: The authors used the words “retracted article” and “retracted manuscript” in most places. For example, on page 9, line 179, the author writes about the subject analysis of retracted manuscripts in China, and on page 9, line 180, the author writes about the subject analysis of retracted articles from Chinese authors. The research is based on retracted articles, not manuscripts.

Response: Thank you very much. We have replaced the term “Retracted manu-scripts” with “Retracted article” and have thoroughly verified this correction in the revised manuscript.

Question 4: Figure 4: Autocorrelation analysis mapping is difficult to under-stand; you can provide a table or provide more explanations.

Response: Thank you very much for your suggestions. We have added mote ex-planations about fig 4 as follows:

 Fig 4 is the result of trend surface analysis. The trend surface analysis is a mul-tivariate statistical method based on the least squares approach, and this method was employed to accurately fit the regional distribution and temporal trends in sample data. By considering the number of retracted articles with Chinese authors across 31 provinces along with the corresponding longitude or latitude of each province, we established a binary polynomial regression model and presented it through a three-dimensional visualization to effectively demonstrate the overall spatial trend of re-traction occurrences. In this study, we used ArcGIS 10.3 to analyze, the result shown in:

 Unfortunately, the results can only be presented in the form of graphs, and the software cannot output the corresponding value. The X-axis and Y-axis represent lon-gitude and latitude, and the Z-axis represents the number of retracted papers in each year. A point (X, Y, Z) in the three-dimensional space represents the number of re-tracted papers in a certain research area. The green line represents the east-west di-rection, and the blue line represents the north-south direction. The steeper the trend line change, the greater the difference.

The results described in Fig4 was presented in the section of results as follows:

The trend surface fitting results indicate that the spatial trend of 31 provinces in China demonstrates an inverted ‘U’ type distribution in the north-south direction from 2012 to 2023. Moreover, there is a noteworthy spatial directivity pattern in the east-west direction, with elevated values in the east and lower values in the west. For instance, in 2023, as longitude increases, the number of retractions also increases; meanwhile, as dimension decreases, the number of retractions initially rises and then declines. High-heat areas for the number of retractions are identified in the eastern and central regions. In addition, the trend line alterations vary in differing directions, with a steeper transition of the trend surface in the east-west direction a

---

## [Decision Letter · Decision Letter 1]

6 Nov 2024

PONE-D-24-27000R1Mapping retracted articles and exploring regional differences in China, 2012-2023PLOS ONE

Dear Dr. He,

Thank you for submitting your manuscript to PLOS ONE. After careful consideration, we feel that it has merit but does not fully meet PLOS ONE’s publication criteria as it currently stands. Therefore, we invite you to submit a revised version of the manuscript that addresses the points raised during the review process.

We look forward to receiving your revised manuscript.

Kind regards,

Robin Haunschild

Academic Editor

PLOS ONE

**Journal Requirements:**

Reviewers' comments:

Reviewer's Responses to Questions

**Comments to the Author**

1. If the authors have adequately addressed your comments raised in a previous round of review and you feel that this manuscript is now acceptable for publication, you may indicate that here to bypass the “Comments to the Author” section, enter your conflict of interest statement in the “Confidential to Editor” section, and submit your "Accept" recommendation.

Reviewer #1: All comments have been addressed

Reviewer #2: All comments have been addressed

Reviewer #3: All comments have been addressed

2. Is the manuscript technically sound, and do the data support the conclusions?

Reviewer #1: Yes

Reviewer #2: Yes

Reviewer #3: Yes

3. Has the statistical analysis been performed appropriately and rigorously? 

Reviewer #1: Yes

Reviewer #2: Yes

Reviewer #3: Yes

4. Have the authors made all data underlying the findings in their manuscript fully available?

Reviewer #1: Yes

Reviewer #2: Yes

Reviewer #3: Yes

5. Is the manuscript presented in an intelligible fashion and written in standard English?

Reviewer #1: Yes

Reviewer #2: No

Reviewer #3: Yes

6. Review Comments to the Author

**Reviewer #1:** I have gone through the revised paper. You are to be congratulated for your research on such a pressing topic. In most retracted or retraction studies China happened to top the list of retraction. Hopefully, this paper will raise the authors', publishers', and policymakers' attention so that they can take the necessary action.

**Reviewer #2:** There are no comments to the author; the author made all the corrections as suggested.

As stated in the review, the author implemented an analysis of prominent institutes, publishers, and authors, adhering to the suggestion to replace the term "retracted manuscripts" with "retracted articles." The author provided additional explanations for Figure 4 and incorporated the most recent references related to retracted articles. The author has implemented the suggestions and corrections mentioned in the review. The paper is recommended for publishing.

**Reviewer #3:** I appreciate the authors for revising the manuscript and submitting the revision. I recommend to accept it. However, I have some minor queries.

1. In page 6, a sentence reads "The Retraction Watch Database was completely open and freely available on September 12, 2023." What is the relation of mentioning the date here? Because it is freely available. In my opinion, the entire paragraph is not required since it is repeated in the next para.

2. A sub-heading reads "Subject, institutes, and publishers Analysis of retracted articles in China" which should be revised.

3. According to authors, the reasons have been categorized into nine and they have referred an previous article (refer page 11). Contrary to this, Table S5 caption reads "Standardized reason categorization corresponding to categorization employed by the Retraction Watch Database". Which is correct?

7. PLOS authors have the option to publish the peer review history of their article (what does this mean?). If published, this will include your full peer review and any attached files.

Reviewer #1: No

Reviewer #2: **Yes: **Dr. N. Siva

Reviewer #3: **Yes: **B. Elango

---

## [Author Response · Author response to Decision Letter 1]

11 Nov 2024

Response to Reviewers

Dear Editor,

We are grateful for the valuable comments and suggestions given by the review-ers concerning our manuscript entitled “Mapping retracted articles and exploring regional differences in China, 2012-2023” (ID: PONE-D-24-27000R1). We give our responses to the reviewer in this letter. We hope that after our replies, our manuscript will be considered for publication in PLOS ONE. The detailed responses are listed below, and revised parts of the manuscript are marked with red color.

Question 1: Please review your reference list to ensure that it is complete and correct. If you have cited papers that have been retracted, please include the ra-tionale for doing so in the manuscript text, or remove these references and replace them with relevant current references. Any changes to the reference list should be mentioned in the rebuttal letter that accompanies your revised manuscript. If you need to cite a retracted article, indicate the article’s retracted status in the Refer-ences list and also include a citation and full reference for the retraction notice.

Response: Thanks. We have carefully reviewed our reference list to ensure that all citations are complete and accurate. For any missing references, we have made the necessary additions, which are highlighted in red in the revised manuscript for your easy reference.

Reviewer #1: 

I have gone through the revised paper. You are to be congratulated for your re-search on such a pressing topic. In most retracted or retraction studies China hap-pened to top the list of retraction. Hopefully, this paper will raise the authors', pub-lishers', and policymakers' attention so that they can take the necessary action.

Response: Thank you very much for your thoughtful review and constructive feedback. We sincerely appreciate your recognition of the importance of this research on such a pressing issue. Your insights and support have been invaluable in enhanc-ing the quality of this work. Once again, thank you for your valuable input.

Reviewer #2:

There are no comments to the author; the author made all the corrections as suggested. As stated in the review, the author implemented an analysis of prominent institutes, publishers, and authors, adhering to the suggestion to replace the term "retracted manuscripts" with "retracted articles." The author provided additional explanations for Figure 4 and incorporated the most recent references related to retracted articles. The author has implemented the suggestions and corrections mentioned in the review. The paper is recommended for publishing.

Response: Thank you for your detailed and constructive feedback. We greatly appreciate your positive evaluation and support, as your feedback has been invalua-ble in helping us refine the paper. Once again, thank you for taking the time and ef-fort to review our manuscript. 

Reviewer #3:

Question 1. In page 6, a sentence reads "The Retraction Watch Database was completely open and freely available on September 12, 2023." What is the relation of mentioning the date here? Because it is freely available. In my opinion, the en-tire paragraph is not required since it is repeated in the next paragraph.

Response: Thanks for your suggestion. We have deleted the entire paragraph in our revised manuscript.

Question 2. A sub-heading reads "Subject, institutes, and publishers Analysis of retracted articles in China" which should be revised.

Response: Thanks for your suggestion. We have revised the subheading in the revised manuscripts, as follows: Analysis of Subjects, Publishers, and Institutions Involved in Retracted Articles in China

Question 3. According to authors, the reasons have been categorized into nine and they have referred an previous article (refer page 11). Contrary to this, Table S5 caption reads "Standardized reason categorization corresponding to categoriza-tion employed by the Retraction Watch Database". Which is correct?

Response: Thank you for your suggestion. In this study, we primarily standard-ized the retraction reasons obtained from the Retraction Watch Database ( for de-tailed reasons, please see: https://retractionwatch.com/retraction-watch-database-user-guide/retraction-watch-database-user-guide-appendix-b-reasons/) into nine dis-tinct categories, as referred to the article https://doi.org/10.1007/s11192-021-03895-1. Since the categorization of retraction reasons in Retraction Watch Database has certain limitations, such as “investigations by company/institution”, “investigation by third party”, “investigation by journal/publisher”, “rogue editor”, which do not reflect actual retraction reasons, we have addressed these issues in our analysis. Ta-ble S5 presents the relationship between these standardized categories and the rea-sons provided by the Retraction Watch Database, for example, the first row outlines the classification of retraction reasons as defined in the Retraction Watch Database, which includes categories such as Plagiarism of Text, Plagiarism of Image, Plagia-rism of Data, Plagiarism of Article, Euphemisms for Plagiarism, these categories are collectively categorized under the broader term 'Plagiarism' , as referenced in the article.

---

## [Editor Report · Decision Letter 2]

14 Nov 2024

Mapping retracted articles and exploring regional differences in China, 2012-2023

PONE-D-24-27000R2

Dear Dr. He,

We’re pleased to inform you that your manuscript has been judged scientifically suitable for publication and will be formally accepted for publication once it meets all outstanding technical requirements.

Kind regards,

Robin Haunschild

Academic Editor

PLOS ONE
---

## [Editor Report · Acceptance letter]

19 Nov 2024

PONE-D-24-27000R2 

PLOS ONE

Dear Dr. He, 

I'm pleased to inform you that your manuscript has been deemed suitable for publication in PLOS ONE. Congratulations! Your manuscript is now being handed over to our production team.

Kind regards, 

on behalf of

Dr. Robin Haunschild 

Academic Editor

PLOS ONE